# Contribution of Ezrin on the Cell Surface Plasma Membrane Localization of Programmed Cell Death Ligand-1 in Human Choriocarcinoma JEG-3 Cells

**DOI:** 10.3390/ph14100963

**Published:** 2021-09-24

**Authors:** Mayuka Tameishi, Takuro Kobori, Chihiro Tanaka, Yoko Urashima, Takuya Ito, Tokio Obata

**Affiliations:** 1Laboratory of Clinical Pharmaceutics, Faculty of Pharmacy, Osaka Ohtani University, Tondabayashi, Osaka 584-8540, Japan; u4117083@osaka-ohtani.ac.jp (M.T.); u4117078@osaka-ohtani.ac.jp (C.T.); urasiyo@osaka-ohtani.ac.jp (Y.U.); 2Laboratory of Natural Medicines, Faculty of Pharmacy, Osaka Ohtani University, Tondabayashi, Osaka 584-8540, Japan; itoutaku@osaka-ohtani.ac.jp

**Keywords:** programmed cell death ligand-1, ezrin/radixin/moesin, gestational trophoblastic neoplasia, immune check point inhibitor

## Abstract

Immune checkpoint blockade (ICB) antibodies targeting programmed cell death ligand-1 (PD-L1) and programmed cell death-1 (PD-1) have improved survival in patients with conventional single agent chemotherapy-resistant gestational trophoblastic neoplasia (GTN). However, many patients are resistant to ICB therapy, the mechanisms of which are poorly understood. Unraveling the regulatory mechanism for PD-L1 expression may provide a new strategy to improve ICB therapy in patients with GTN. Here, we investigated whether the ezrin/radixin/moesin (ERM) family, i.e., a group of scaffold proteins that crosslink actin cytoskeletons with several plasma membrane proteins, plays a role in the regulation of PD-L1 expression using JEG-3 cells, a representative human choriocarcinoma cell line. Our results demonstrate mRNA and protein expressions of ezrin, radixin, and PD-L1, as well as their colocalization in the plasma membrane. Intriguingly, immunoprecipitation experiments revealed that PD-L1 interacted with both ezrin and radixin and the actin cytoskeleton. Moreover, gene silencing of ezrin but not radixin strongly diminished the cell surface expression of PD-L1 without altering the mRNA level. These results indicate that ezrin may contribute to the cell surface localization of PD-L1 as a scaffold protein in JEG-3 cells, highlighting a potential therapeutic target to improve the current ICB therapy in GTN.

## 1. Introduction

Gestational trophoblastic neoplasia (GTN) is a pregnancy-related unique malignant lesion arising from placental villous and extravillous trophoblast [1]. GTN consists of four clinicopathologic conditions: the malignant invasive mole, choriocarcinoma, rare placental site trophoblastic tumors (PSTT), and epithelioid trophoblastic tumors (ETT) [1,2]. Globally, approximately 18,000 women are diagnosed with GTN annually, most of whom are cured with single agent chemotherapy [3,4]. However, 0.5–5.0% of women die due to the development of multidrug resistance, highlighting the need for a novel therapeutic approach to GTN [3].

Immune checkpoint proteins, such as programmed cell death-1 (PD-1) and programmed cell death ligand-1 (PD-L1), are molecules that negatively regulate the activation of T cell immune response. Once PD-L1, expressed on cell surface membranes of various cancers and macrophages in tumor tissue [5,6,7,8], binds to PD-1 on the activated cytotoxic T cells, immune checkpoint signaling pathways are activated, and these shut down antitumor T cell immunity [9]. Intriguingly, evidence has indicated that PD-L1 is also strongly expressed in GTN [10,11,12,13], suggesting a crucial role of PD-L1 in the escape of GTN from the host immune response by T cells, which leads to immune tolerance.

In recent years, therapeutic antibodies (Abs) against PD-L1 (e.g., atezolizumab, avelumab, and durvalumab) and PD-1 (e.g., nivolumab, pembrolizumab, spartalizumab, and cemiplimab) have been shown to reactivate T cell immunity in the tumor microenvironment, leading to the elimination of tumor cells and remarkable survival benefits in various advanced cancers [14,15,16,17]. Additionally, recent clinical studies have impressively shown that the use of Abs against PD-1 and PD-L1 markedly reverses trophoblastic tolerance in patients with single agent chemotherapy-resistant GTN [3,18,19]. Unfortunately, less than 40% of patients show clinical benefits because of their primary and adaptive resistance to PD-1/PD-L1 blockade therapies [20,21,22,23,24,25], the mechanisms of which are largely unknown.

The protein expression of PD-L1 is intricately regulated by various cellular processes, such as gene transcription, post-transcriptional and post-translational modifications, and exosomal transport [20,26,27]. Since PD-L1 is a transmembrane protein, emerging evidence has demonstrated that the protein expression levels of PD-L1 in the plasma membrane are considerably regulated by post-translational modifications, such as phosphorylation, glycosylation, ubiquitination, and palmitoylation, which affect the localization and functional activity of PD-L1 [20,26,27,28,29]. Therefore, exploring the druggable potential regulators for PD-L1 expression in the surface plasma membrane may help to improve the current immune checkpoint blockade (ICB) therapies.

Members of the ezrin/radixin/moesin (ERM) family of proteins crosslink the actin cytoskeleton and various plasma membrane proteins, such as several drug transporters, including P-glycoprotein (P-gp), multidrug resistant protein (MRP)-2, and MRP-3 [30,31,32], and other cancer-related proteins, including epidermal growth factor receptor 2 [33]. Interestingly, gene silencing of moesin dramatically decreased the plasma membrane localization of PD-L1 in human breast cancer cell lines, indicating a novel regulatory mechanism of PD-L1 by ERM family proteins via post-translational modifications [34]. However, it is unclear whether ERM also regulates the plasma membrane localization of PD-L1 in other cancer cell types.

In this study, we aimed to determine the expression profile and cellular localization of PD-L1 and ERM, in addition to examining the role of ERM in the cell surface localization of PD-L1 by gene silencing methods and immuno-precipitation experiments in JEG-3 cells, a representative human choriocarcinoma cell line.

## 2. Results

### 2.1. Gene and Protein Expression Profiles of PD-L1 and Each ERM in JEG-3 Cells

Expression levels of ezrin, radixin, moesin, and PD-L1 mRNA in JEG-3 cells were measured using real-time quantitative reverse transcription-polymerase chain reaction (RT-PCR) analysis. The mRNA expression levels of ezrin, radixin, and PD-L1 were sufficient to allow gene expression analysis (Figure 1a–c). While the mRNA expression of moesin was detected at a higher level in primary human umbilical vein endothelial cells (HUVEC) and HeLa cells, a human uterine cervix cell line, both have been used as positive control cells, in which moesin is abundantly present [35,36,37,38], while moesin levels were considerably lower in JEG-3 cells (Figure 1d, Appendix A). Similarly, protein expression levels of PD-L1, in addition to ezrin and radixin, were detected in whole cell lysates of JEG-3 cells (Figure 1e). The moesin protein was undetectable in whole cell lysates of JEG-3 cells, but strongly detected in those of HeLa cells (Figure 1e). Therefore, moesin was excluded from subsequent analysis. Next, we analyzed the gene expression profiles of PD-L1 and ERM in three human choriocarcinoma cell lines (JEG-3, JAR, and T3M-3 cells) registered in the public database of the Cancer Cell Line Encyclopedia (CCLE) [39] and the Cancer Dependency Map (DepMap) portal data explorer [40,41]. The database analysis revealed that the mRNA expressions of PD-L1, ezrin, and radixn were abundant in JEG-3 cells, and their relative expression levels were intermediate between JAR and T3M-3 cells. In contrast, T3M-3 cells, but not JEG-3 cells and JAR, carry gene encoding moesin, which is in agreement with our present results (Appendix A).

### 2.2. Subcellular Localization of Ezrin, Radixin, and PD-L1 in JEG-3 Cells

The intracellular distribution of ezrin, radixin, and PD-L1 in JEG-3 cells was confirmed using fluorescent immunostaining. Ezrin and radixin both displayed cellular distribution in JEG-3 cells similar to that of β-actin, a plasma membrane marker, indicating that ezrin and radixin are localized in the plasma membrane (Figure 2a,b). PD-L1 was co-localized with actin but not with the nuclear marker, demonstrating that PD-L1 is preferentially present in the plasma membrane (Figure 2c,d). Importantly, PD-L1 was co-localized strongly with ezrin and radixin in the plasma membrane (Figure 3a,b).

### 2.3. PD-L1 Interacts with Ezrin and Radixin in JEG-3 Cells

We next determined whether PD-L1 interacts with ezrin and radixin in JEG-3 cells. The protein expressions of ezrin, radixin, and PD-L1, as well as β-actin, were all detected in immunoprecipitates from whole cell fractions of JEG-3 cells pulled down using an anti-PD-L1 Ab (Figure 4). These results suggest the protein–protein interaction of PD-L1 with ezrin and radixin, as well as the actin cytoskeleton in JEG-3 cells.

### 2.4. Effect of siRNAs against Ezrin and Radixin, Respectively, on Expression Levels of Target mRNAs and Proteins in JEG-3 Cells

Small interfering RNAs (siRNAs) for ezrin and radixin were transfected into JEG-3 cells to suppress the target gene and protein levels. siRNAs for ezrin and radixin significantly suppressed the expression of each target mRNA by 70–90% compared with Lipofectamine, a transfection reagent, alone, without affecting the expression of another gene (Figure 5a,b). However, 5 nM nontargeting control (NC) siRNA treatment increased the mRNA levels of ezrin and radixin, and 2 nM NC siRNA treatment also increased ezrin levels (Figure 5a,b). Additionally, siRNA targeting ezrin and radixin also significantly decreased the expression levels of the respective target protein by 75–90% compared with Lipofectamine alone (Figure 5c). We also confirmed that cell viability was not decreased by any of the siRNA treatments used in this study (Figure 5d).

### 2.5. Effect of ERM Silencing on mRNA and Cell Surface Expressions of PD-L1 in JEG-3 Cells

PD-L1 mRNA levels were unchanged by siRNAs targeting ezrin and radixin (Figure 6a). Note that the suppressive effect of siRNA for ezrin on the target mRNA and protein expression levels was higher than that for radixin, although both siRNAs significantly decreased protein levels and each target mRNA. Additionally, the gene silencing of PD-L1 with siRNA significantly reduced its target mRNA levels (Figure 6b). We finally determined whether the gene silencing of ezrin and radixin affects the cell surface expressions of PD-L1, which enabled us to identify the involvement of ezrin and radixin in the expression of PD-L1 at the post-translational level. The results of flow cytometry analysis showed that the gene silencing of ezrin, but not of radixin, significantly suppressed the protein expression levels of PD-L1 on the cell surface to the same level as the gene silencing of PD-L1 (Figure 6c,d). These results indicate that ezrin contributed to the plasma membrane localization of PD-L1 in JEG-3 cells.

## 3. Discussion

In the present study, we first detected the higher expression levels of PD-L1 together with ezrin and radixin, while also detecting considerably lower levels and a deficit of moesin at the genetic and protein levels in JEG-3 cells, respectively. The results of these observations are in accordance with those of previous reports that demonstrated an abundance of ezrin and radixin, in contrast to a lack of moesin, at the protein level in JEG-3 cells [38,42]. Therefore, moesin was excluded from subsequent analysis. Notably, we discovered for the first time that JEG-3 cells are rich in PD-L1 expression at both the genetic and protein levels. Using counterstaining with plasma membrane and nuclear markers, confocal laser scanning microscopy (CLSM) analysis showed that PD-L1, ezrin, and radixin are preferentially distributed in the plasma membrane, but not in the nuclei, in JEG-3 cells. Interestingly, PD-L1 was highly colocalized with ezrin and radixin in the plasma membrane. Although there is a lack of data on in vitro cell lines derived from human gestational trophoblastic tumors, several histopathological studies have demonstrated the strong expression of PD-L1 in the plasma membrane of tumor cells obtained from human gestational trophoblastic neoplasia [10,12,13]. In addition, Higuchi et al. immunohistochemically identified that ezrin is highly localized in the apical membranes of a syncytiotrophoblast cell line, which is a source of GTN derived from rat placenta [43]. Taken together, these findings suggest that PD-L1, ezrin, and radixin, but not moesin, are highly expressed at the genetic and protein levels in JEG-3 cells, and that PD-L1 is co-localized with ezrin and radixin in the plasma membrane of this cell line.

Accumulated evidence suggests that the protein expression level and functional activity of plasma membrane proteins are not always dependent on the transcriptional process [30,44,45,46]. Recently, two research groups independently identified the previously obscured chemokine-like factor-like MARVEL transmembrane domain containing 6 (CMTM6) as a crucial regulator of PD-L1 in a wide range of cancer cells, and discovered through a genome-wide screen method that CMTM6 binds PD-L1 and contributes to cell surface localization by preventing PD-L1 from being targeted for lysosomal degradation in various cancer cell types [28,29]. They also demonstrated that CMTM6 increases the PD-L1 protein pool without influencing PD-L1 transcription levels, and rather that CMTM6 interacts with the PD-L1 protein at the cell surface and reduces its ubiquitination, leading to a prolonged half-life of the PD-L1 protein [28]. ERM proteins have been recognized as crucial factors for several drug transporters involved in multi-drug resistance and cancer-related plasma membrane proteins by retaining these proteins in the plasma membrane of cancer cells via post-translational modifications [30,31,32,33,47]. Additionally, Ghosh et al. reported that phosphorylated ERM family proteins colocalize with T cell receptor (TCR) αβ, a member of the immunoglobulin (IgG) superfamily of proteins, as well as with actin filaments, suggesting a novel function of ERM in crosslinking the TCR complex to the actin cytoskeleton [48]. More recently, Meng et al. found that moesin interacts and colocalizes with PD-L1, and that the phosphorylation of moesin is necessary for the stabilization of PD-L1 on the cell surface membrane in human breast cancer cell lines [34]. Our present immunoprecipitation analysis provides novel evidence that both ezrin and radixin physiologically interact with PD-L1. Therefore, these present and previous findings allow us to hypothesize that ezrin and radixin may serve as scaffold proteins regulating the plasma membrane localization of PD-L1 via post-translational modifications in JEG-3 cells.

To determine the roles of ezrin and radixin in the gene and/or cell surface protein expression levels of PD-L1, we adopted RNAi methods to induce gene silencing of ezrin and radixin in JEG-3 cells. We confirmed that each siRNA against ezrin and radixin strongly and selectively decreased the respective target mRNA and protein expression levels with few impacts on cytotoxicity. Thus, we succeeded in developing an in vitro experimental model to determine the roles of ezrin and radixin in the gene and/or protein expressions of PD-L1 in JEG-3 cells. While siRNA targeting ezrin and radixin both exhibited no impact on the gene expression levels of PD-L1, the knockdown of ezrin, but not of radixin, significantly suppressed the protein expression levels of PD-L1 in the cell surface plasma membrane of JEG-3 cells, although the knockdown activity of siRNA for ezrin on the target mRNA and protein expression levels was higher than that for radixin. Meng et al. have also shown that cell surface PD-L1 levels were dramatically suppressed by the gene silencing of moesin without any impact on its mRNA expression level, resulting in T cell activation by an in vitro cell culture model, although the effect of ezrin and radixin gene suppression has yet to be determined [34]. Since ERM proteins involved in the plasma membrane localization of P-gp (a well-recognized ERM partner protein) differ according to cancer type, organ, and animal species [30,49,50,51,52,53], this discrepancy among the present and previous results may be at least in part due to the different expression profiles of ERM in cancer cell types; however, the details remain unclear. Taken together, these observations suggest that ezrin contributes to the plasma membrane localization of PD-L1, possibly as a scaffold protein, by crosslinking PD-L1 with the actin cytoskeleton in JEG-3 cells (Figure 7). Based on these observations, the agents inhibiting ezrin expression may help to improve the current ICB therapies, possibly by modulating the cell surface expression of the PD-L1 protein in human choriocarcinoma cells.

## 4. Materials and Methods

### 4.1. Cell Culture

The human gestational choriocarcinoma cell line, JEG-3, was purchased from the European Collection of Cell Cultures (ECACC) collections (EC92120308-F0; KAC, Hyogo, Japan). JEG-3 cells were cultured in Dulbecco’s modified Eagle medium (DMEM) containing 1500 mg/L glucose (FUJIFILM Wako Pure Chemical, Osaka, Japan) supplemented with heat-inactivated 10% fetal bovine serum (FBS) (BioWest, Nuaillé, France). The cultures were maintained at 37 °C in a humidified atmosphere with 5% CO_2_.

### 4.2. siRNA Treatment

JEG-3 cells were cultured overnight to allow for attachment at a density of 2.0 × 10^4^ cells/well in 24-well cell culture plates (Corning, Glendale, AZ, USA) for extraction of total RNA and flow cytometry analysis, at 8.0 × 10^4^ cells/well in 6-well cell culture plates (Corning) for total protein isolation, and at 4.0 × 10^3^ cells/well in 96-well cell culture plates (Corning) for cell viability assays. Then, cells were transfected with Silencer Select siRNAs (Thermo Fisher Scientific, Tokyo, Japan) targeting human ezrin or PD-L (2 nM), and that targeting human radixin (5 nM) diluted with Opti-MEM (Thermo Fisher Scientific) using the Lipofectamine RNAiMAX Transfection Reagent (Thermo Fisher Scientific). Silencer Select Negative Control siRNA (Thermo Fisher Scientific) was used as a negative control for each siRNA. The volume of transfection reagent used was 0.20 µL/well for total RNA isolation and flow cytometry analysis, 0.80 µL/well for total protein isolation, and 0.04 µL/well for cell viability assays.

### 4.3. Extraction of Total RNA and Real-Time Reverse Transcription-Polymerase Chain Reaction (RT-PCR)

ISOSPIN Cell & Tissue RNA (NIPPON GENE, Tokyo, Japan) was used for total RNA isolation, according to the manufacturer’s protocol, followed by a measurement of the quality and quantity of total RNA using a NanoDrop ND-1000 spectrophotometer (Thermo Fisher Scientific). The mRNA expression levels of each target gene were determined by RT-PCR, as described previously [53,54], with some modifications. The sequences of gene-specific primers (all purchased from Takara Bio) are shown in Table 1. The relative fold changes in the mRNA levels of each target gene normalized to that of the internal control, β-actin, were calculated with the comparative quantification cycle (Cq) method (2^−ΔΔCq^).

### 4.4. Confocal Laser Scanning Microscopy (CLSM) Analysis

CLSM analysis was conducted as described previously, with some modifications [53,54,55].

#### 4.4.1. Single Immunofluorescence Staining

JEG-3 cells were plated on a polylysine-coated 35 mm glass bottom dish (Matsunami Glass, Osaka, Japan) at a density of 0.5–1.0 × 10^5^ cells/dish and cultured overnight to allow for cell attachment. Then, cells were washed with Dulbecco’s phosphate-buffered saline (D-PBS) (FUJIFILM Wako Pure Chemical) and fixed with 4% paraformaldehyde (PFA) (FUJIFILM Wako Pure Chemical) for 15 min at room temperature, followed by washing with D-PBS thrice. After that, the cells were permeabilized with 0.5% Triton-X100 (Thermo Fisher Scientific) for 15 min at room temperature, followed by washing with D-PBS three times. Subsequently, cells were blocked in a blocking buffer consisting of D-PBS supplemented with 1% bovine serum albumin (BSA) (FUJIFILM Wako Pure Chemical), 10% normal goat serum (Thermo Fisher Scientific), 0.3 M glycine (FUJIFILM Wako Pure Chemical), and 0.1% Tween-20 (Nacalai Tesque, Kyoto, Japan) for 60 min at room temperature to avoid non-specific protein–protein interactions. In the experiments to observe the cellular localization of ezrin and radixin, cells were reacted overnight at 4 °C with a mouse anti-β-actin antibody (Ab) (A1978; Merck) at a dilution of 1:20 in combination with a rabbit anti-ezrin Ab (3145S; Cell Signaling Technology, Danvers, MA, USA) at a dilution of 1:50, or a rabbit anti-radixin Ab (GTX105408; GeneTex, Alton Pkwy Irvine, CA, USA) at a dilution of 1:50. After rinsing them with D-PBS supplemented with 0.1% Tween-20 (PBS-T) three times, the cells were reacted for 60 min at room temperature with an Alexa Fluor 488-conjugated donkey anti-mouse IgG (H+L) Highly Cross-Adsorbed secondary Ab (A-21202; Thermo Fisher Scientific) at a dilution of 1:500 for β-actin or an Alexa Fluor 594-conjugated goat anti-rabbit IgG (H+L) Cross-Adsorbed Ready Probes secondary Ab (R37117; Thermo Fisher Scientific) at a dilution of 1 drop/500 µL for ezrin and radixin. The cells were then rinsed thrice with PBS-T and sealed with a drop of Fluoro-KEEPER Antifade Reagent containing 4′,6-diamidine-2′-phenylindole dihydrochloride (DAPI) (Nacalai Tesque) to counterstain nuclei and preserve fluorescence for 30 min at room temperature. Thereafter, photomicrographs were taken at 0.3–0.5 µm intervals for the z-axis at an original magnification of ×60–120 using a Nikon A1 confocal laser microscope system (Nikon Instruments, Tokyo, Japan). The two- or three-dimensional images were reconstructed from the acquired data using NIS-Elements Ar Analysis software (Nikon Instruments).

In the localization analysis of PD-L1 using nuclear and plasma membrane markers, the same procedure was conducted as described above before Ab reactions. Thereafter, the cells were incubated overnight with an Alexa Fluor 488-conjugated rabbit anti-human PD-L1 Ab (25048; Cell Signaling Technology) at a dilution of 1:50 at 4 °C. After washing them in PBS-T three times, the cells were incubated for 30 min at room temperature with a blocking buffer containing NucRed Live 647 Ready Probes Reagent (R37106; Thermo Fisher Scientific) at a dilution of 1 drop/500 µL for nuclear counterstaining, or ActinRed 555 Ready Probes Reagent (rhodamine phalloidin) (R37112; Thermo Fisher Scientific) at a dilution of 1 drop/500 µL for cell membrane counterstaining. Subsequently, cells were washed thrice with PBS-T, followed by addition with a drop of Fluorescence Mounting Medium (Agilent, Santa Clara, CA, USA) to preserve fluorescence. After that, photomicrographs were taken as described above.

#### 4.4.2. Double Immunofluorescence Staining

The same procedure was conducted as described above before Ab reactions. Subsequently, cells were reacted overnight at 4 °C with a rabbit anti-ezrin Ab (3145S; Cell Signaling Technology) at a dilution of 1:50, or a rabbit anti- radixin Ab (GTX105408; GeneTex) at a dilution of 1:50. After rinsing them with PBS-T three times, the cells were reacted for 60 min at room temperature with an Alexa Fluor 594-conjugated goat anti-rabbit IgG (H+L) Cross-Adsorbed Ready Probes Secondary Ab (R37117; Thermo Fisher Scientific) at a dilution of 1 drop/500 µL for ezrin and radixin. The cells were then washed three times in PBS-T and incubated overnight with an Alexa Fluor 488-conjugated rabbit anti-human PD-L1 Ab (25048; Cell Signaling Technology) at a dilution of 1:50 in blocking buffer at 4 °C under moist and dark conditions. The cells were then washed thrice again with PBS-T and sealed with a drop of Fluoro-KEEPER Antifade Reagent, Non-Hardening Type (Nacalai Tesque) to preserve fluorescence. Thereafter, photomicrographs were taken as described above.

### 4.5. Cell Viability Assay

Cell viability assay was performed as described previously [53,56,57,58,59], with some modifications. Briefly, three days after the treatment of the cells with siRNAs or staurosporine (10 µM/well, Merck, Darmstadt, Germany), a positive control for inducing cell death, a new resazurin-based PrestoBlue Cell Viability Reagent (Invitrogen), was added to culture medium and incubated for 10 min. Thereafter, fluorescence signals were measured at a wavelength of 560 nm (excitation) and 590 (emission) and 10 nm (bandwidth) using a Synergy HTX Multi-Mode Microplate Reader (Bio Tek Instrument, Winooski, VT, USA).

### 4.6. Protein Isolation

After the treatment of cells with siRNAs or staurosporine for three days without changing the medium, cells were rinsed twice with ice-cold D-PBS and subsequently lysed in radio-immunoprecipitation assay (RIPA) buffer containing protease inhibitor cocktails for 30 min on ice. The cell debris were removed via centrifugation (15,000× *g*, 4 °C, 10 min), and the resultant supernatant was collected as the total cell lysate. The protein concentration was quantified using a TaKaRa BCA Protein Assay Kit (Takara Bio).

### 4.7. Western Blotting

Western blotting was conducted as described previously, with some modifications [54,55]. Briefly, total lysates of HeLa cells were diluted with an equal volume of a Sample Buffer Solution (2×) for sodium dodecyl sulfate (SDS)-polyacrylamide gel electrophoresis (PAGE), which consisted of 0.125 M Tris-HCl, 4% SDS, 20% glycerin, 0.01% bromophenol blue, and 10% 2-mercaptoethanol (Nacalai Tesque), then boiled at 97 °C for 5 min. Total protein concentrations (ranging from 5.0 µg/lane to 7.0 µg/lane, dependent on the target proteins) were loaded and separated by SDS-PAGE, followed by their transfer onto a nitrocellulose membrane (Bio-Rad Laboratories) via electrophoresis. Consistent blotting was determined using Ponceau S (MP Biomedicals, Santa Ana, CA, USA) staining. The membrane was incubated in blocking buffer containing 5% non-fat dry milk (FUJIFILM Wako Pure Chemical) in PBS-T for 60 min at room temperature. Subsequently, the membrane was probed with rabbit Abs against ezrin (3145s; Cell Signaling Technology) at a dilution of 1:1000, radixin (GTX105408; Gene Tex) at a dilution of 1:2000, or moesin (3150s; Cell Signaling Technology) at a dilution of 1:1000, then probed with a mouse Ab against glyceraldehyde-3-phosphate dehydrogenase (GAPDH) (MAB374; Merck) at a dilution of 1:20,000 as an internal control, or horse radish peroxidase (HRP)-conjugated rabbit Ab against PD-L1 (51296s; Cell Signaling Technology) at a dilution of 1:2000 at 4 °C overnight. Blots were then washed with PBS-T and incubated with a HRP-conjugated secondary Ab against rabbit IgG (5220-0336; SeraCare Life Sciences, Milford, MA, USA) at a dilution of 1:5000 for ezrin, radixin, and moesin, or against mouse IgG (5220-0341; SeraCare Life Sciences) at a dilution of 1:10,000 for GAPDH at room temperature for 60 min, followed by rinsing with PBS-T. After that, the immunoreactive bands were visualized by a Pierce ECL Western Blotting Substrate (Thermo Fisher Scientific). The chemiluminescence signal intensities of the immune reactive bands were measured using a Light Capture (ATTO, Tokyo, Japan) and analyzed with an Image Analysis Software CS Analyzer (ATTO). All the original western blotting images are shown in Appendix A.

### 4.8. Immunoprecipitation Assay

Immunoprecipitation experiments were conducted as described previously [60,61], with some modifications. Briefly, 500 μL of the total whole cell lysate, prepared in the same way as described above, was incubated with 50 μL of nProtein A Sepharose 4 Fast flow (Cytiva, Tokyo, Japan) for 60 min at 4 °C on a rotating wheel to remove non-specific binding proteins to nProtein A Sepharose. After nProtein A Sepharose was pelleted via centrifugation (3000× *g*, 4 °C for 1 min), the pre-cleaned supernatants of the whole cell lysates were incubated overnight at 4 °C on a rotating wheel with a rabbit Ab against PD-L1 (13684s; Cell Signaling Technology) or its isotype control Ab (3900s; Cell Signaling Technology), both at a dilution of 1:30. Then, 50 μL of nProtein A Sepharose was added into the lysate and subsequently incubated at 4 °C for 3 h on a rotating wheel. The precipitates were rinsed three times with RIPA buffer containing protease inhibitor cocktails, followed by centrifugation (3000× *g*, 4 °C for 1 min) to obtain the immunoprecipitated pellets. After resuspension of the immunoprecipitated pellets in a Sample Buffer Solution (2×) for SDS-PAGE (Nacalai Tesque), the pellets were boiled at 97 °C for 5 min and pelleted by centrifugation (15,000× *g*, 4 °C for 1 min). The supernatant fractions and total cell lysates were adjusted to a protein concentration ranging from 0.7 µg/lane to 7.0 µg/lane, depending on the target proteins, and were loaded and separated via SDS-PAGE, followed by their transfer onto a nitrocellulose membrane (Bio-Rad Laboratories) via electrophoresis. Consistent blotting was determined using Ponceau S (MP Biomedicals) staining. The membrane was incubated in blocking buffer containing 5% non-fat dry milk (FUJIFILM Wako Pure Chemical) in PBS-T for 60 min at room temperature. Subsequently, the membrane was probed with rabbit Abs against ezrin (3145s; Cell Signaling Technology) at a dilution of 1:1000, radixin (GTX105408; Gene Tex) at a dilution of 1:2000, mouse Ab against β-actin (A1978; Merck) at a dilution of 1:10,000, or HRP-conjugated rabbit Ab against PD-L1 (51296s; Cell Signaling Technology) at a dilution of 1:1000 in blocking buffer at 4 °C overnight. Blots were then washed with PBS-T and incubated with HRP-conjugated secondary Abs against a rabbit IgG (5220-0336; SeraCare Life Sciences) at a dilution of 1:5000 for ezrin and radixin, or a mouse IgG (5220-0341; SeraCare Life Sciences) at a dilution of 1:10,000 for β-actin in blocking buffer for 60 min at room temperature. After washing with PBS-T, immune-complexes were visualized using a Pierce ECL Western Blotting Substrate (Thermo Fisher Scientific). The chemiluminescence signal intensities of the immune reactive bands were detected and analyzed using a Light Capture (ATTO) equipped with an Image Analysis Software CS Analyzer (ATTO). All the original western blotting images are shown in Appendix A.

### 4.9. Flow Cytometry Analysis

Flow cytometry analysis was carried out as described previously, with some mod-ifications [53,54,55]. Three days after the incubation of JEG-3 cells with siRNAs, the cells were desquamated with an Accutase (Nacalai Tesque) and rinsed with a labeling buffer consisting of D-PBS supplemented with 5% normal horse serum (Biowest) and 1% sodium azide (FUJIFILM Wako Pure Chemical). After centrifugation (260× *g* for 5 min at 4 °C), the cells were reacted with an allophycocyanin (APC)-conjugated mouse anti-human CD274 (B7-H1, PD-L1) Ab (329708; BioLegend, San Diego, CA, USA) at a dose of 4.0 μg/tube in a labeling buffer for 60 min at 4 °C. After washing the cells with the labeling buffer, followed by centrifugation (260× *g* for 5 min at 4 °C), the precipitated cells were resuspended in D-PBS containing propidium iodide (PI) (Dojindo Laboratories, Kumamoto, Japan) to exclude PI-positive dead cells. Thereafter, cell surface expression levels of PD-L1 were determined using a Cell Analyzer EC800 (Sony Imaging Products & Solutions, Tokyo, Japan). Data were analyzed using the EC800 Analysis software (Sony Imaging Products & Solutions) to calculate the mean fluorescence intensity of the APC-PD-L1 in JEG-3 cells.

### 4.10. Statistical Analysis

All data are expressed as the mean ± standard error of the mean (SEM). Differences in the mean values between the control and treatment groups were assessed by a one-way analysis of variance (ANOVA) with Dunnett’s post-hoc test. GraphPad Prism version 3 software (GraphPad Software, La Jolla, CA, USA) was used for statistical analysis. Differences with *p* values less than 0.05 were considered significant.

## 5. Conclusions

The present study demonstrates that ezrin and radixin, but not moesin, are expressed in JEG-3 cells at the mRNA and protein levels, and are specifically localized in the plasma membrane where both ezrin and radixin are highly colocalized and interact with PD-L1. We also demonstrated that the gene silencing of ezrin, but not radixin, strongly suppressed the cell surface expression of PD-L1 with no influence on the transcriptional level of PD-L1. This indicates a novel role of ezrin in the plasma membrane localization of PD-L1, which may serve as a scaffold protein. Thus, specific inhibition of ezrin may provide a novel strategy to improve the current ICB therapy, possibly by modulating the cell surface expression of PD-L1 in human choriocarcinoma cells.

One limitation of the present study is that the in vitro relationship between PD-L1 and ERM proteins in JEG-3 cells cannot fully mimic clinical patients with choriocarcinoma-received ICB therapy. In addition, the data obtained in this study represent only one type of human choriocarcinoma cell line, despite the existence of genetic and/or phenotypic features that vary from cell lines to cell lines. We should address these issues with more in vitro and in vivo experiments, in addition to developing more clinical evidence to better understand the clinical relationship between PD-L1 and ERM proteins in patients with choriocarcinoma.

## Figures and Tables

**Figure 1 pharmaceuticals-14-00963-f001:**
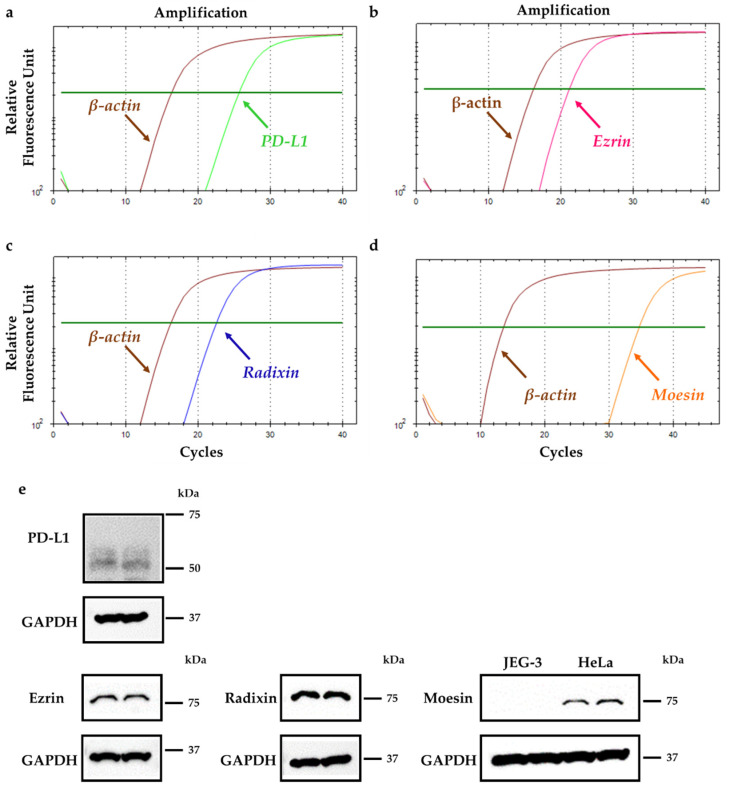
Gene and protein expression patterns of programmed cell death ligand-1 (PD-L1) and ezrin, radixin, and moesin (ERM) in JEG-3 cells. (**a**–**d**) Representative amplification curves of (**a**) PD-L1, (**b**) ezrin, (**c**) radixin, and (**d**) moesin together with β-actin (internal control) in JEG-3 cells as determined using real-time quantitative reverse transcription (RT)-polymerase chain reaction (PCR). (**e**) Typical western blotting images of PD-L1 with ezrin, radixin, and moesin, as well as glyceraldehyde-3-phosphate dehydrogenase (GAPDH) in whole cell lysates of JEG-3 cells. HeLa cells were used as positive controls as they abundantly express moesin protein. Molecular weights are indicated in kDa.

**Figure 2 pharmaceuticals-14-00963-f002:**
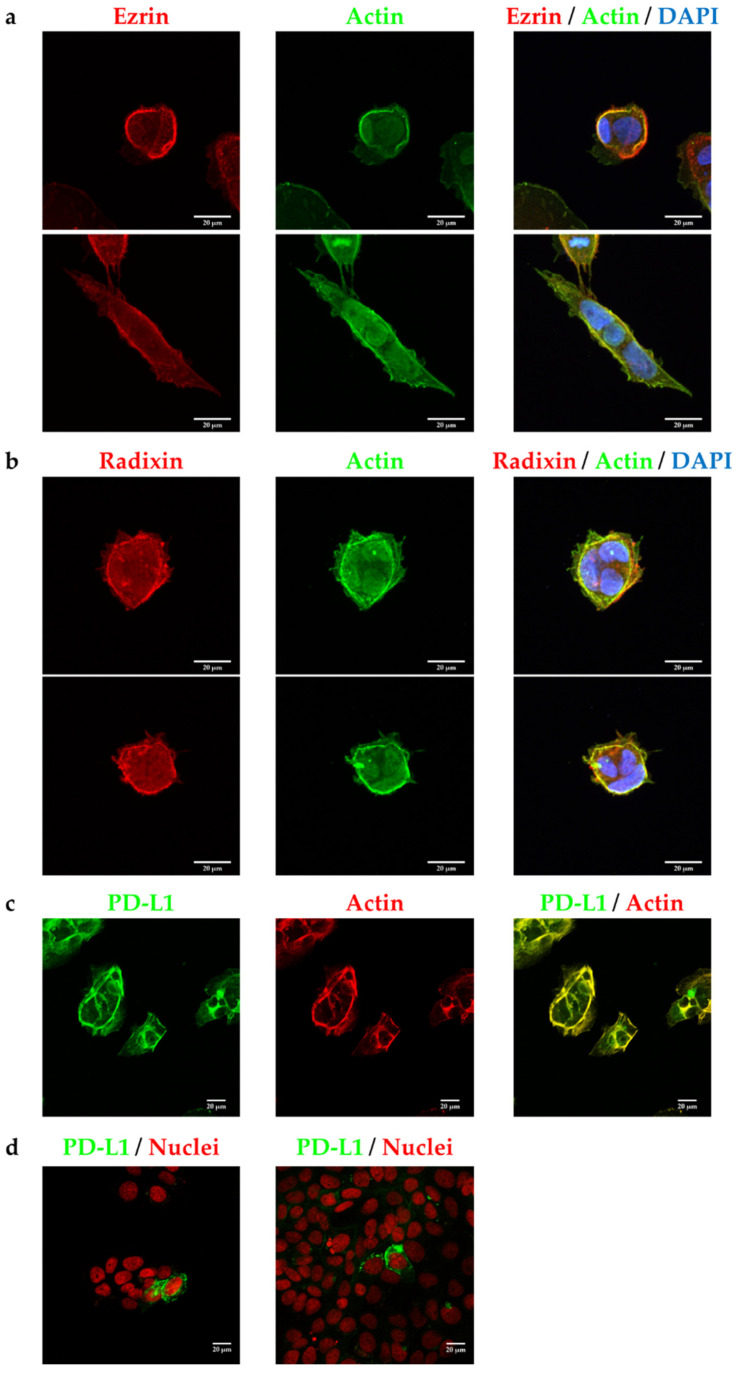
Subcellular localization of ezrin, radixin, and programmed cell death ligand-1 (PD-L1) in JEG-3 cells. Confocal laser scanning microscopy analysis for intracellular distribution of ezrin, radixin, and PD-L1 in JEG-3. In a three-dimensional reconstruction of optically sectioned JEG-3 cells (**a**,**b**), ezrin and radixin (red) were localized at the plasma membrane and preferentially colocalized with actin (green). (**c**,**d**) PD-L1 (green) was localized at the plasma membrane and preferentially colocalized with actin (red) on the plasma membrane, but not with nuclei (red). Scale bars: 20 μm. All data are representative of at least three independent experiments.

**Figure 3 pharmaceuticals-14-00963-f003:**
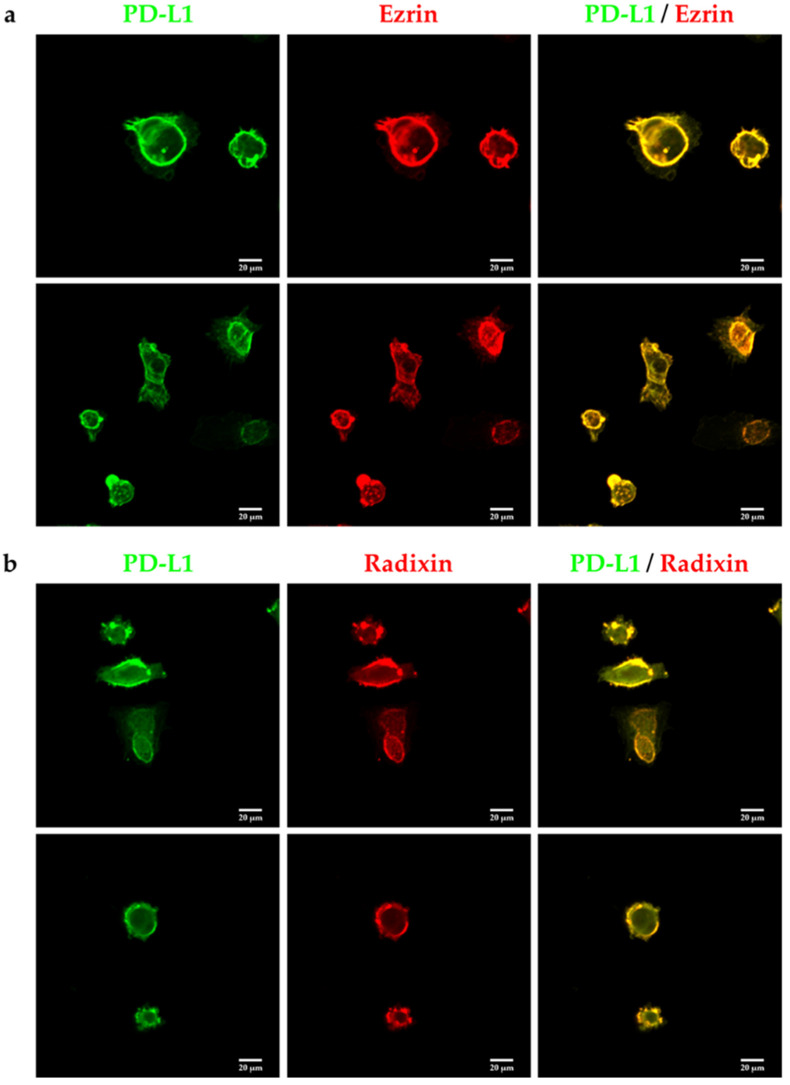
Double immunofluorescence staining analysis of programmed cell death ligand-1 (PD-L1) with ezrin and radixin in JEG-3 cells. Confocal laser scanning microscopy analysis for intracellular distribution of ezrin, radixin, and PD-L1 in JEG-3 cells. PD-L1 (green) was localized at the plasma membrane and highly colocalized with both (**a**) ezrin and (**b**) radixin (red). Scale bars: 20 μm. All data are representative of at least three independent experiments.

**Figure 4 pharmaceuticals-14-00963-f004:**
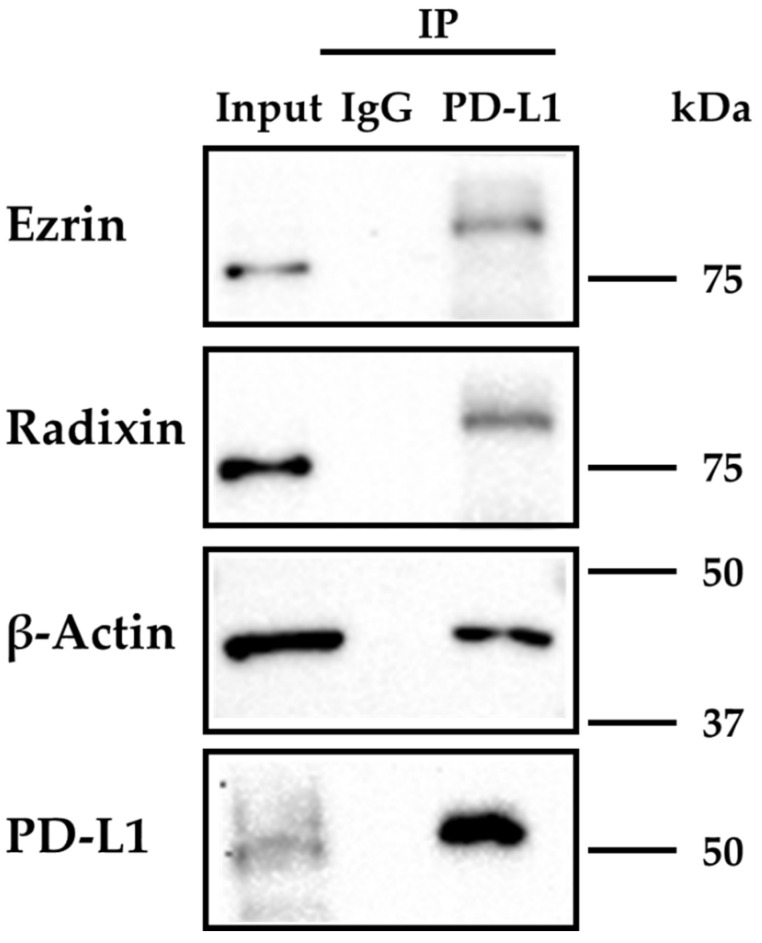
Immunoprecipitation analysis to detect protein–protein interactions of programmed cell death ligand-1 (PD-L1) with ezrin and radixin in JEG-3 cells. The whole cell lysates of JEG-3 cells were immunoprecipitated with an anti-PD-L1 antibody or its isotype-matched control antibody. Typical western blotting images of ezrin, radixin, and PD-L1, as well as β-actin, in whole cell lysates (input) and those in immunoprecipitates (IP) using a control antibody or an anti-PD-L1 antibody. Molecular weights are indicated in kDa.

**Figure 5 pharmaceuticals-14-00963-f005:**
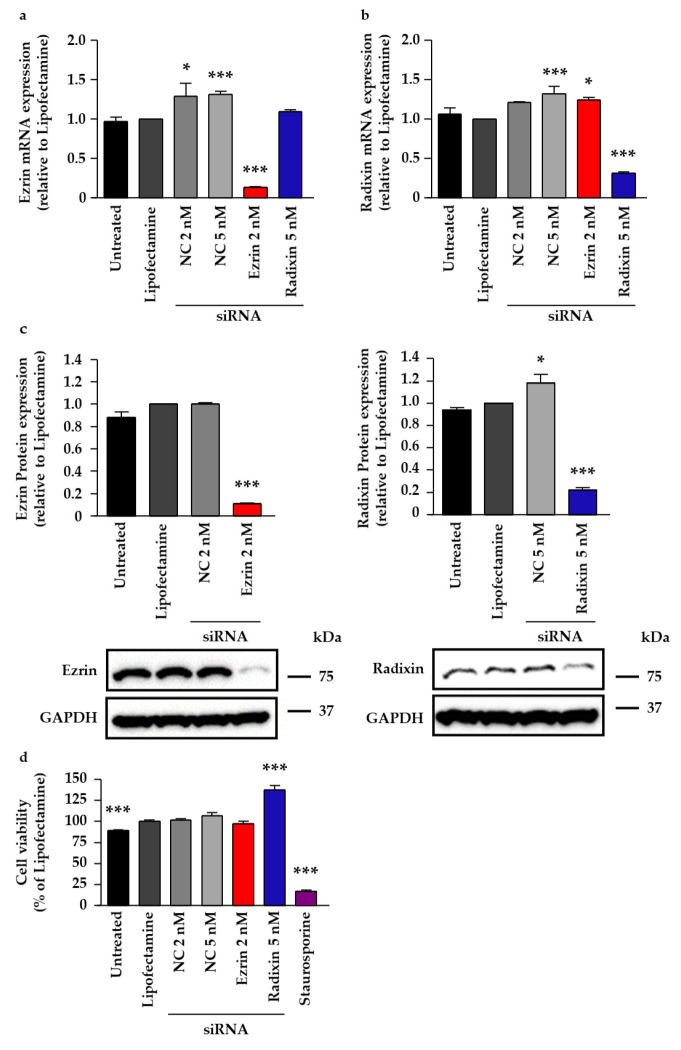
Effect of siRNAs targeting ezrin or radixin on the respective mRNA and protein expression levels and on the cell viability of JEG-3 cells. Cells were treated with the transfection medium (Untreated), transfection reagent (Lipofectamine), nontargeting control (NC) siRNA, and specific siRNAs for ezrin or radixin and then incubated for three days. (**a**,**b**) Expression levels of each mRNA normalized to β-actin in the cells treated with siRNAs relative to that in cells treated with the transfection reagent alone were measured using real-time quantitative reverse transcription-polymerase chain reaction. *n* = 3–5, *** *p* < 0.001, * *p* < 0.05 vs. Lipofectamine. (**c**) Expression levels of each protein normalized to glyceraldehyde-3-phosphate dehydrogenase (GAPDH) in the cells treated with siRNAs relative to that in cells treated with the transfection reagent alone were measured using western blotting analysis. Typical blotting images of ezrin and radixin, as well as GAPDH, in whole cell lysates of JEG-3 cells. Molecular weights are indicated in kDa. *n* = 3, *** *p* < 0.001, * *p* < 0.05 vs. Lipofectamine. (**d**) Cell viability was assessed using PrestoBlue cell viability reagent. Staurosporine was used as a positive control for inducing cell death. *n* = 7–15, *** *p* < 0.001 vs. Lipofectamine. All data are expressed as the mean ± SEM and were analyzed using one-way ANOVA, followed by Dunnett’s test.

**Figure 6 pharmaceuticals-14-00963-f006:**
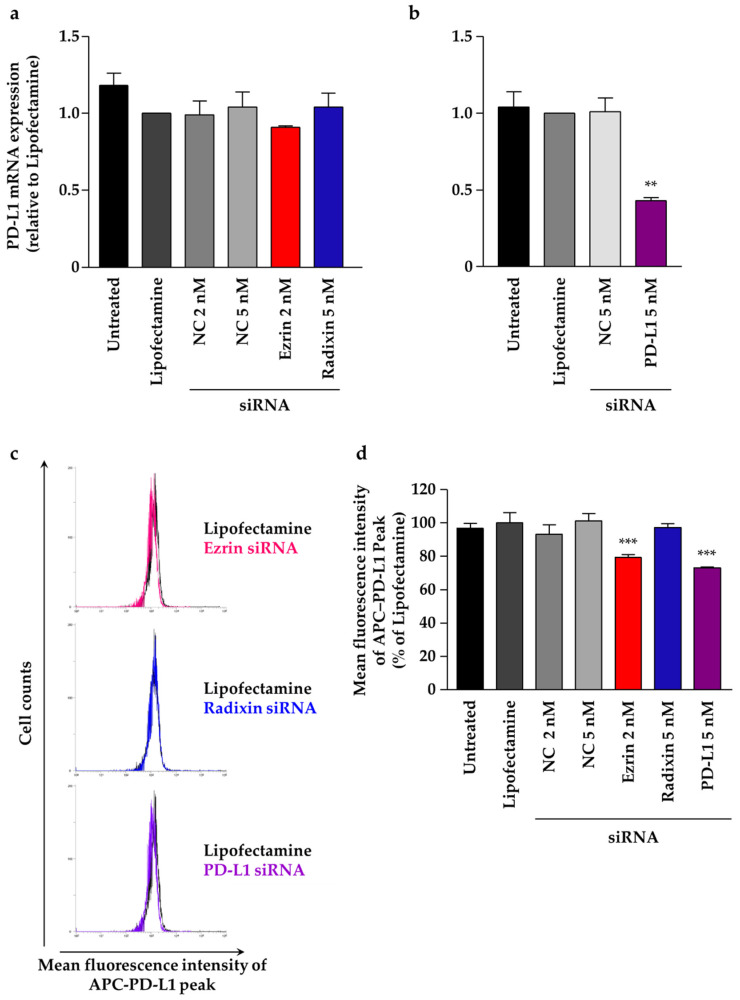
Effect of siRNAs targeting ezrin, radixin, or programmed cell death ligand-1 (PD-L1) on mRNA and cell surface expressions of PD-L1 in JEG-3 cells. Cells were treated with the transfection medium (Untreated), transfection reagent (Lipofectamine), nontargeting control (NC) siRNA (2 nM or 5 nM), and specific siRNAs for ezrin (2 nM), radixin (5 nM), or PD-L1 (5 nM), and then incubated for three days. The expression level of PD-L1 mRNA normalized to β-actin in cells treated with each siRNA relative to that in cells treated with Lipofectamine alone was determined using real-time quantitative reverse transcription-polymerase chain reaction. (**a**) *n* = 3–5, (**b**) *n* = 3, ** *p* < 0.01 vs. Lipofectamine. (**c**) An overlay of the representative histograms for the mean fluorescence intensity of allophycocyanin (APC)-labeled PD-L1 on the surface plasma membrane in JEG-3 cells treated with Lipofectamine (black line), ezrin siRNA (red line), radixin siRNA (blue line), and PD-L1 siRNA (purple line), as measured using flow cytometry. (**d**) The calculated mean fluorescence intensities of PD-L1 relative to Lipofectamine alone on the plasma membrane surface are shown for all the treatments; *n* = 3, *** *p* < 0.001 vs. Lipofectamine. All data were expressed as the mean ± SEM and analyzed using one-way ANOVA, followed by Dunnett’s test.

**Figure 7 pharmaceuticals-14-00963-f007:**
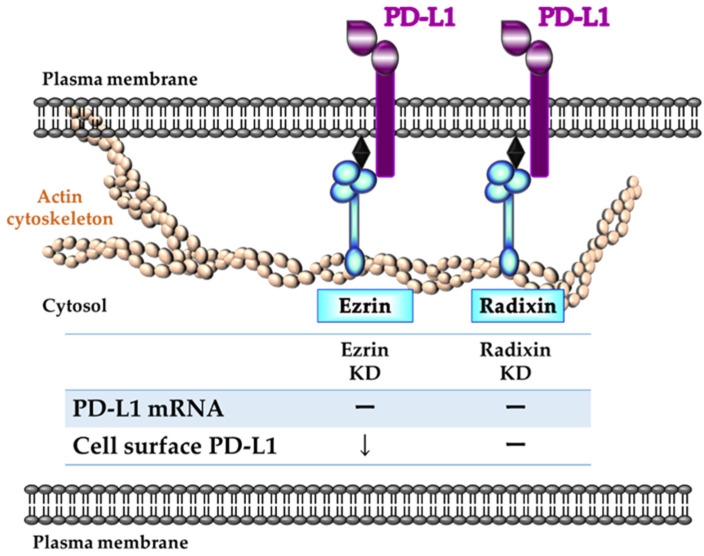
Schematic diagram of a proposed model illustrating the different functions of ezrin and radixin in the regulatory mechanism of programmed cell death ligand-1 (PD-L1) expression in JEG-3 cells. Ezrin contributes to the cell surface expression of PD-L1, possibly via protein–protein interactions, without influencing its mRNA. Despite the presence of protein–protein interactions between PD-L1 and radixin, radixin may have no impact on either mRNA or the cell surface expression levels of PD-L1. Therefore, among ERM proteins, ezrin may play an essential role as a scaffold protein, contributing to the cell surface plasma membrane localization of PD-L1 in JEG-3 cells. Arrow indicates the reduction of PD-L1 expression.

**Table 1 pharmaceuticals-14-00963-t001:** Primer sequences used in this study.

Gene	Primer Sequence (5′→3′)
*h-β-actin* (forward)	TGGCACCCAGCACAATGAA
*h*-*β-actin* (reverse)	CTAAGTCATAGTCCGCCTAGAAGCA
*h*-*ezrin* (forward)	ACCATGGATGCAGAGCTGGAG
*h*-*ezrin* (reverse)	CATAGTGGAGGCCAAAGTACCACA
*h*-*radixin* (forward)	GAATTTGCCATTCAGCCCAATA
*h*-*radixin* (reverse)	GCCATGTAGAATAACCTTTGCTGTC
*h*-*moesin* (forward)	CCGAATCCAAGCCGTGTGTA
*h*-*moesin* (reverse)	GGCAAACTCCAGCTCTGCATC
*h*-*PD-L1* (forward)	CAATGTGACCAGCACACTGAGAA
*h*-*PD-L1* (reverse)	GGCATAATAAGATGGCTCCCAGAA

## Data Availability

The datasets used and analyzed during this study are available from Cancer Cell Line Encyclopedia (https://portals. broadinstitute.org/ccle/) (accessed on 1 September 2021) and Cellular Models Expression (https://depmap.org/portal/interactive) (accessed on 1 September 2021). Other data are contained within the article and Appendix A.

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
