# Peer review of "Contribution of Ezrin on the Cell Surface Plasma Membrane Localization of Programmed Cell Death Ligand-1 in Human Choriocarcinoma JEG-3 Cells"

_pharmaceuticals, 2021, doi:10.3390/ph14100963_

Round 1
Reviewer 1 Report
This paper reports interesting results on the effects of the ezrin/radixin/moesin (ERM) family proteins crosslinking the actin cytoskeleton with various membrane proteins on the cell surface plasma membrane localization of programmed cell death ligand-1 (PD-L1) in human choriocarcinoma cells. However, there are some issues concerning their presentation and discussion that should be addressed in order to improve the final version of the paper as it is itemized below.
Specific points
- “Results” section - Figure 5c showing typical western blotting images obtained in the study should be completed with quantitative data based on densitometry analysis to confirm the statement of the Authors that siRNA targeting ezrin and radixin also greatly decreased respective target protein expression levels (lines 145-146 and lines 246-247).
- The careful revision of the statements found in lines 170-173 and 250-253 as well as in the Legend to Figure 9 that the knockdown of ezrin but not radixin significantly suppressed the protein expression levels of PD-L1 should be done, since the western blotting images presented in Figure 5c indicate that siRNA gene silencing decreased the respective protein product levels in much lesser extent in the case of radixin than in the case of ezrin.
Minor points
- The expression: “the inhibitory effect of the target gene level” (line 140) requires some revision.
- Lines 145-146 - “siRNA 5 nM” should be replaced by “5 nM siRNA” and “siRNA 2 nM” should be replaced by “2 nM siRNA”.
- It is recommended to change the colors used in Figure 6c for better distinguishing the individual histograms presented in Figure 6c.
Author Response
Response to Reviewer 1 Comments
We would like to thank #Reviewer 1 for the valuable suggestions and constructive comments on our manuscript. We have carefully read all of your suggestions and comments, and have made the corrections in the revised version of manuscript. Detailed responses to your comments are listed below, and we highlighted all changes with word track changes in the file labeled ‘Revised Manuscript with Track Changes’. We hope this revised manuscript would be satisfactory for publication in Pharmaceuticals.
Specific Points
Comment 1. “Results section” - Figure 5c showing typical western blotting images obtained in the study should be completed with quantitative data based on densitometry analysis to confirm the statement of the Authors that siRNA targeting ezrin and radixin also greatly decreased respective target protein expression levels (lines 145-146 and lines 246-247).
Reply Comments.
We would like to appreciate #Reviewer 1’s valuable suggestion. As #Reviewer 1 pointed out, we have incorporated the quantitative data based on the densitometry analysis in Figure 5c. As a result, siRNAs targeting ezrin and radixin both significantly suppressed each target protein expression level in comparison with control. According to the addition of these results, we have also revised the sentence in the Result section as follows.
Results (Line 151 – 152)
Additionally, siRNA targeting ezrin and radixin also significantly decreased the ex-pression levels of respective target protein by 70–90% compared with Lipofectamine alone (Figure 5c).
Legend for Figure 5 (Line 164 – 168)
(c) Expression levels of each protein normalized to glyceraldehyde-3-phosphate dehydrogenase (GAPDH) in the cells treated with siRNAs relative to that in cells treated with the transfection reagent alone were measured using western blotting analysis. Typical blotting images of ezrin and radixin, as well as GAPDH in whole cell lysates of JEG-3 cells. Molecular weights are indicated in kDa. n = 3, ***p < 0.001, *p < 0.05 vs. Lipofectamine.
Comment 2. The careful revision of the statements found in lines 170-173 and 250-253 as well as in the Legend to Figure 6 that the knockdown of ezrin but not radixin significantly suppressed the protein expression levels of PD-L1 should be done, since the western blotting images presented in Figure 5c indicate that siRNA gene silencing decreased the respective protein product levels in much lesser extent in the case of radixin than in the case of ezrin.
Reply Comments.
We would like to appreciate #Reviewer 1’s valuable suggestion. According to the additional results that siRNA against ezrin and radixin significantly suppressed each target protein expression level in the Specific Comment 1, we replaced the western blotting image of radixin with other more typical one in Figure 5c. However, as #Reviewer 1 pointed out, the suppressive effect of target protein product levels is different between ezrin and radixin siRNAs, although a significant decrease in respective target protein levels were observed in both siRNAs. Therefore, we have incorporated the notification to explain the difference in the suppressive effects of target proteins between ezrin and radixin siRNAs into the Results and Discussion section.
Results (Line 151 – 152)
Additionally, siRNA targeting ezrin and radixin also significantly decreased the ex-pression levels of respective target protein by 70–90% compared with Lipofectamine alone (Figure 5c).
Results (Line 174 – 176)
Note that the suppressive effect of siRNA for ezrin on the target mRNA and protein expression levels was higher than that for radixin, although both siRNAs significantly decreased each target mRNA and protein levels.
Discussion (Line 259 – 263)
While siRNA targeting ezrin and radixin both exhibited no impact on gene expression levels of PD-L1, the knockdown of ezrin but not radixin significantly suppressed the protein expression levels of PD-L1 in the cell surface plasma membrane of JEG-3 cells, although the knockdown activity of siRNA for ezrin on the target mRNA and protein expression levels was higher than that for radixin.
Minor Points
Comment 1. The expression: “the inhibitory effect of the target gene level” (line 140) requires some revision.
Reply Comments.
We would like to appreciate #Reviewer 1’s valuable suggestion. As #Reviewer 1 pointed out, we have revised the sentence as follows.
Results (Line 145 – 146)
Small interfering RNAs (siRNAs) for ezrin and radixin were transfected into JEG-3 cells to decrease the target gene and proteins levels.
Comment 2. Lines 145-146 - “siRNA 5 nM” should be replaced by “5 nM siRNA” and “siRNA 2 nM” should be replaced by “2 nM siRNA”.
Reply Comments.
We would like to appreciate #Reviewer 1’s prompt check. As #Reviewer 1 pointed out, we have replaced siRNA X nM with X nM siRNA as follows.
Results (Line 149 – 151)
However, 5 nM nontargeting control (NC) siRNA treatment increased the mRNA levels of ezrin and radixin, and 2 nM NC siRNA treatment also increased ezrin levels (Figure 5a-b).
Comment 3. It is recommended to change the colors used in Figure 6c for better distinguishing the individual histograms presented in Figure 6c.
Reply Comments.
We would like to appreciate #Reviewer 1’s valuable suggestion. As #Reviewer 1 pointed out, to enhance the understanding of this Figure easily for many readers, we have replaced the combined histogram with the individual ones all of which compare each siRNA with Lipofectamine control in Figure 6c.
Legends for Figure 6 (Line 193 – 196)
(c) An overlay of the representative histograms for the mean fluorescence intensity of allophycocyanin (APC)-labeled PD-L1 on the surface plasma membrane in JEG-3 cells treated with Lipofectamine (black line), ezrin siRNA (red line), radixin siRNA (blue line), and PD-L1 siRNA (purple line), as measured using flow cytometry.

Reviewer 2 Report
Reviewer’s Comments:
It has been my pleasure to review the article by Mayuka Tameishi and colleagues, titled “Contribution of Ezrin on the Cell Surface Plasma Membrane 2 Localization of Programmed Cell Death Ligand-1 in Human Choriocarcinoma JEG-3 Cells”. The authors discuss the interaction of ezrin and radixin in JEG-3 cells at mRNA and protein levels and are specifically localized in the plasma membrane where both ezrin and radixin are highly colocalized and interact with PD-L1. Thus, providing a novel strategy that can implement to improve the current ICB therapy by specific inhibition of ezrin. I have read the article with significant interest, and I think that such a review should be considered for publication, although major changes are necessary.
Major Comments:
1) The current study demonstrates ezrin and radixin in JEG-3 cells at mRNA and protein levels, and are specifically localized in the plasma membrane where both ezrin and radixin are highly colocalized and interact with PD-L1. Do other Human Choriocarcinoma cell lines such as BeWo and SV63 can show a similar effect. Please demonstrate this effect of 3 different Choriocarcinoma cell lines and a non-Choriocarcinoma cell line as a negative control to a wider picture of the interaction of ezrin and radixin and its colocalization and interaction with PD-L1?
2) Please Provide a full blot for all the western Blot data?
3) Erzin and Radixin localization in plasma membrane and interaction with PD-L1, thus specific inhibition of ezrin can be a novel strategy to improve ICB therapy. Please provide in-vitro and in-vivo experimental data to support this statement?
Author Response
Response to Reviewer 2 Comments
We would like to thank #Reviewer 2 for the valuable suggestions and constructive comments on our manuscript. We have carefully read all of your suggestions and comments, and have made the corrections in the revised version of manuscript. Detailed responses to your comments are listed below, and we highlighted all changes with word track changes in the file labeled ‘Revised Manuscript with Track Changes’. We hope this revised manuscript would be satisfactory for publication in Pharmaceuticals.
Major Comments
Comment 1. The current study demonstrates ezrin and radixin in JEG-3 cells at mRNA and protein levels, and are specifically localized in the plasma membrane where both ezrin and radixin are highly colocalized and interact with PD-L1. Do other Human Choriocarcinoma cell lines such as BeWo and SV63 can show a similar effect. Please demonstrate this effect of 3 different Choriocarcinoma cell lines and a non-Choriocarcinoma cell line as a negative control to a wider picture of the interaction of ezrin and radixin and its colocalization and interaction with PD-L1?
Reply Comments.
We would like to appreciate #Reviewer 2’s valuable suggestion. As #Reviewer 2 pointed out, we also think it is extremely important to determine whether the colocalization and the molecular interaction between ezrin/radixin and PD-L1 is exist in other human choriocarcinoma cell lines to generalize our concept. In fact, to rapidly expand our idea for other types of cancer cells, we have recently demonstrated the role of ERM proteins in the cell surface localization of PD-L1 in LS180 cells, a human colorectal cancer cell line (Kobori, T. et al., Pharmaceuticals 2021, 14(9), 864; https://doi.org/10.3390/ph14090864) and HeLa cells, a human cervical cancer cells (Tanaka, C. et al., Molecules, under revised process). Therefore, we used and focused JEG-3 cells having abundant expression level of PD-L1 with ezrin and radixin in this manuscript. However, based on the valuable opinions from #Reviewer 2, we mentioned the limitation of this study in the Conclusion section. Furthermore, to check the gene expression profiles of PD-L1 and ERM in some human choriocarcinoma cell lines, we analyzed the relative gene expression patterns in three human choriocarcinoma cell lines (JEG-3, JAR, and T3M-3 cells) registered in the public database of the Cancer Cell Line Encyclopedia (CCLE) and the Cancer Dependency Map (DepMap) portal data explorer. The obtained results were shown in Figure S2.
Again, thank you for your valuable suggestions and constructive comments.
Results (Line 93 – 100)
Next, we analyzed the gene expression profiles of PD-L1 and ERM in three human choriocarcinoma cell lines (JEG-3, JAR, and T3M-3 cells) registered in the public database of the Cancer Cell Line Encyclopedia (CCLE) [39] and the Cancer Dependency Map (DepMap) portal data explorer [40, 41]. The database analysis revealed that the mRNA expressions of PD-L1, ezrin, and radixn were abundant in JEG-3 cells, and their relative expression levels were intermediate between JAR and T3M-3 cells. In contrast, T3M-3 cells but not JEG-3 cells and JAR carry gene encoding moesin that is in agree-ment with our present results (Figure S2).
References (Line 606 – 622)
- Barretina, J.; Caponigro, G.; Stransky, N.; Venkatesan, K.; Margolin, A. A.; Kim, S.; Wilson, C. J.; Lehar, J.; Kryukov, G. V.; Sonkin, D.; Reddy, A.; Liu, M.; Murray, L.; Berger, M. F.; Monahan, J. E.; Morais, P.; Meltzer, J.; Korejwa, A.; Jane-Valbuena, J.; Mapa, F. A.; Thibault, J.; Bric-Furlong, E.; Raman, P.; Shipway, A.; Engels, I. H.; Cheng, J.; Yu, G. K.; Yu, J.; Aspesi, P., Jr.; de Silva, M.; Jagtap, K.; Jones, M. D.; Wang, L.; Hatton, C.; Palescandolo, E.; Gupta, S.; Mahan, S.; Sougnez, C.; Onofrio, R. C.; Liefeld, T.; MacConaill, L.; Winckler, W.; Reich, M.; Li, N.; Mesirov, J. P.; Gabriel, S. B.; Getz, G.; Ardlie, K.; Chan, V.; Myer, V. E.; Weber, B. L.; Porter, J.; Warmuth, M.; Finan, P.; Harris, J. L.; Meyerson, M.; Golub, T. R.; Morrissey, M. P.; Sellers, W. R.; Schlegel, R.; Garraway, L. A., The Cancer Cell Line Encyclopedia enables predictive modelling of anticancer drug sensitivity. Nature 2012, 483, 603-607.
- Meyers, R. M.; Bryan, J. G.; McFarland, J. M.; Weir, B. A.; Sizemore, A. E.; Xu, H.; Dharia, N. V.; Montgomery, P. G.; Cowley, G. S.; Pantel, S.; Goodale, A.; Lee, Y.; Ali, L. D.; Jiang, G.; Lubonja, R.; Harrington, W. F.; Strickland, M.; Wu, T.; Hawes, D. C.; Zhivich, V. A.; Wyatt, M. R.; Kalani, Z.; Chang, J. J.; Okamoto, M.; Stegmaier, K.; Golub, T. R.; Boehm, J. S.; Vazquez, F.; Root, D. E.; Hahn, W. C.; Tsherniak, A., Computational correction of copy number effect improves specificity of CRISPR-Cas9 essentiality screens in cancer cells. Nat. Genet. 2017, 49, 1779-1784.
- Tsherniak, A.; Vazquez, F.; Montgomery, P. G.; Weir, B. A.; Kryukov, G.; Cowley, G. S.; Gill, S.; Harrington, W. F.; Pantel, S.; Krill-Burger, J. M.; Meyers, R. M.; Ali, L.; Goodale, A.; Lee, Y.; Jiang, G.; Hsiao, J.; Gerath, W. F. J.; Howell, S.; Merkel, E.; Ghandi, M.; Garraway, L. A.; Root, D. E.; Golub, T. R.; Boehm, J. S.; Hahn, W. C., Defining a Cancer Dependency Map. Cell 2017, 170, 564-576 e516.
Conclusions (Line 474 – 481)
The limitation of the present study is that the in vitro relationship between PD-L1 and ERM proteins in JEG-3 cells cannot fully mimic the clinical patients with choriocarcinoma received ICB therapy. In addition, the data obtained in this study represents only one type of human choriocarcinoma cell line, despite the existence of genetic and/or phenotypic feature varied from cell lines to cell lines. We should address these issues with more in vitro and in vivo experiments in addition to develop more clinical evidence for better understanding the clinical relationship between PD-L1 and ERM proteins in patients with choriocarcinoma.
Supplementary Materials (Line 30 – 38)
Gene expression profile of programmed cell death ligand-1 (PD-L1), ezrin, radixin, and moesin in human choriocarcinoma cell lines.
Figure S2. Gene expression profile of programmed cell death ligand-1 (PD-L1), ezrin, radixin, and moesin in human choriocarcinoma cell lines. Relative gene expression patterns of PD-L1 in addition to ezrin, radixin, and moesin in three human choriocarcinoma cell lines, JAR, JEG-3, and T3M-3, registered in the database of Cancer Cell Line Encyclopedia (CCLE) were determined by utilizing the Cancer Dependency Map (DepMap) portal data explorer. Scatter plots showing the expression levels (log2 (TPM+1)) of each gene in human choriocarcinoma cell lines. Data from CCLE and DepMap were obtained from the 2021Q3 release.
Comment 2. Please Provide a full blot for all the western Blot data?
Reply Comments.
We would like to appreciate #Reviewer 2’s comment. We have incorporated the original western blotting membrane data as Figure S3 in the Supplementary Materials.
Supplementary Materials (Line 40 – 49)
Original Western Blotting images of programmed cell death ligand-1 (PD-L1), ezrin, radixin, and moesin as well as glyceralde-hyde-3-phosphate dehydrogenase (GAPDH) in JEG-3 cells.
Figure S3. Original Western Blotting images of programmed cell death ligand-1 (PD-L1), ezrin, radixin, and moesin as well as glyceraldehyde-3-phosphate dehydrogenase (GAPDH) in JEG-3 cells. (a) The original Western Blotting membrane to confirm the protein expression of PD-L1, ezrin, radixin, and moesin shown in Figure 1e. (b) The original Western Blotting membrane to detect the protein-protein interaction between PD-L1 and ezrin, radixin as well as actin in the whole cell lysates (input) and those in the immunoprecipitates (IP) using a control antibody or an anti-PD-L1 antibody shown in Figure 4. (c) The original Western Blotting membrane to measure the protein expression levels of ezrin and radixin as well as GAPDH shown in Figure 5c.
Comment 3. Ezrin and Radixin localization in plasma membrane and interaction with PD-L1, thus specific inhibition of ezrin can be a novel strategy to improve ICB therapy. Please provide in-vitro and in-vivo experimental data to support this statement?
Reply Comments.
We would like to appreciate #Reviewer 2’s valuable suggestion. As #Reviewer 2 pointed out, in vitro and in vivo experimental data are absolutely essential to support our statement that specific inhibition of ezrin can be a novel strategy to improve ICB therapy. However, there is no convincing evidence that ezrin function as a scaffold protein to stabilize the plasma membrane localization of PD-L1, leading to the resistance to the current ICB therapy in patients with any cancers. Therefore, we should address these issues with more and more in vitro and in vivo experiments in future studies. With a huge emphasis on the valuable opinions from #Reviewer 2, we mentioned the limitation of this study in the Conclusion section.
Again, we would like to thank your valuable suggestions and constructive comments.
Conclusions (Line 474 – 481)
The limitation of the present study is that the in vitro relationship between PD-L1 and ERM proteins in JEG-3 cells cannot fully mimic the clinical patients with choriocarcinoma received ICB therapy. In addition, the data obtained in this study represents only one type of human choriocarcinoma cell line, despite the existence of genetic and/or phenotypic feature varied from cell lines to cell lines. We should address these issues with more in vitro and in vivo experiments in addition to develop more clinical evidence for better understanding the clinical relationship between PD-L1 and ERM proteins in patients with choriocarcinoma.
Round 2
Reviewer 2 Report
Author's Notes
Response to Reviewer 2 Comments
We would like to thank #Reviewer 2 for the valuable suggestions and constructive comments on our manuscript. We have carefully read all of your suggestions and comments, and have made the corrections in the revised version of manuscript. Detailed responses to your comments are listed below, and we highlighted all changes with word track changes in the file labeled ‘Revised Manuscript with Track Changes’. We hope this revised manuscript would be satisfactory for publication in Pharmaceuticals.
Major Comments
Comment 1. The current study demonstrates ezrin and radixin in JEG-3 cells at mRNA and protein levels, and are specifically localized in the plasma membrane where both ezrin and radixin are highly colocalized and interact with PD-L1. Do other Human Choriocarcinoma cell lines such as BeWo and SV63 can show a similar effect. Please demonstrate this effect of 3 different Choriocarcinoma cell lines and a non-Choriocarcinoma cell line as a negative control to a wider picture of the interaction of ezrin and radixin and its colocalization and interaction with PD-L1?
Reply Comments.
We would like to appreciate #Reviewer 2’s valuable suggestion. As #Reviewer 2 pointed out, we also think it is extremely important to determine whether the colocalization and the molecular interaction between ezrin/radixin and PD-L1 is exist in other human choriocarcinoma cell lines to generalize our concept. In fact, to rapidly expand our idea for other types of cancer cells, we have recently demonstrated the role of ERM proteins in the cell surface localization of PD-L1 in LS180 cells, a human colorectal cancer cell line (Kobori, T. et al., Pharmaceuticals 2021, 14(9), 864; https://doi.org/10.3390/ph14090864) and HeLa cells, a human cervical cancer cells (Tanaka, C. et al., Molecules, under revised process). Therefore, we used and focused JEG-3 cells having abundant expression level of PD-L1 with ezrin and radixin in this manuscript. However, based on the valuable opinions from #Reviewer 2, we mentioned the limitation of this study in the Conclusion section. Furthermore, to check the gene expression profiles of PD-L1 and ERM in some human choriocarcinoma cell lines, we analyzed the relative gene expression patterns in three human choriocarcinoma cell lines (JEG-3, JAR, and T3M-3 cells) registered in the public database of the Cancer Cell Line Encyclopedia (CCLE) and the Cancer Dependency Map (DepMap) portal data explorer. The obtained results were shown in Figure S2.
Again, thank you for your valuable suggestions and constructive comments.
Results (Line 93 – 100)
Next, we analyzed the gene expression profiles of PD-L1 and ERM in three human choriocarcinoma cell lines (JEG-3, JAR, and T3M-3 cells) registered in the public database of the Cancer Cell Line Encyclopedia (CCLE) [39] and the Cancer Dependency Map (DepMap) portal data explorer [40, 41]. The database analysis revealed that the mRNA expressions of PD-L1, ezrin, and radixn were abundant in JEG-3 cells, and their relative expression levels were intermediate between JAR and T3M-3 cells. In contrast, T3M-3 cells but not JEG-3 cells and JAR carry gene encoding moesin that is in agree-ment with our present results (Figure S2).
References (Line 606 – 622)
- Barretina, J.; Caponigro, G.; Stransky, N.; Venkatesan, K.; Margolin, A. A.; Kim, S.; Wilson, C. J.; Lehar, J.; Kryukov, G. V.; Sonkin, D.; Reddy, A.; Liu, M.; Murray, L.; Berger, M. F.; Monahan, J. E.; Morais, P.; Meltzer, J.; Korejwa, A.; Jane-Valbuena, J.; Mapa, F. A.; Thibault, J.; Bric-Furlong, E.; Raman, P.; Shipway, A.; Engels, I. H.; Cheng, J.; Yu, G. K.; Yu, J.; Aspesi, P., Jr.; de Silva, M.; Jagtap, K.; Jones, M. D.; Wang, L.; Hatton, C.; Palescandolo, E.; Gupta, S.; Mahan, S.; Sougnez, C.; Onofrio, R. C.; Liefeld, T.; MacConaill, L.; Winckler, W.; Reich, M.; Li, N.; Mesirov, J. P.; Gabriel, S. B.; Getz, G.; Ardlie, K.; Chan, V.; Myer, V. E.; Weber, B. L.; Porter, J.; Warmuth, M.; Finan, P.; Harris, J. L.; Meyerson, M.; Golub, T. R.; Morrissey, M. P.; Sellers, W. R.; Schlegel, R.; Garraway, L. A., The Cancer Cell Line Encyclopedia enables predictive modelling of anticancer drug sensitivity. Nature 2012, 483, 603-607.
- Meyers, R. M.; Bryan, J. G.; McFarland, J. M.; Weir, B. A.; Sizemore, A. E.; Xu, H.; Dharia, N. V.; Montgomery, P. G.; Cowley, G. S.; Pantel, S.; Goodale, A.; Lee, Y.; Ali, L. D.; Jiang, G.; Lubonja, R.; Harrington, W. F.; Strickland, M.; Wu, T.; Hawes, D. C.; Zhivich, V. A.; Wyatt, M. R.; Kalani, Z.; Chang, J. J.; Okamoto, M.; Stegmaier, K.; Golub, T. R.; Boehm, J. S.; Vazquez, F.; Root, D. E.; Hahn, W. C.; Tsherniak, A., Computational correction of copy number effect improves specificity of CRISPR-Cas9 essentiality screens in cancer cells. Nat. Genet. 2017, 49, 1779-1784.
- Tsherniak, A.; Vazquez, F.; Montgomery, P. G.; Weir, B. A.; Kryukov, G.; Cowley, G. S.; Gill, S.; Harrington, W. F.; Pantel, S.; Krill-Burger, J. M.; Meyers, R. M.; Ali, L.; Goodale, A.; Lee, Y.; Jiang, G.; Hsiao, J.; Gerath, W. F. J.; Howell, S.; Merkel, E.; Ghandi, M.; Garraway, L. A.; Root, D. E.; Golub, T. R.; Boehm, J. S.; Hahn, W. C., Defining a Cancer Dependency Map. Cell 2017, 170, 564-576 e516.
Conclusions (Line 474 – 481)
The limitation of the present study is that the in vitro relationship between PD-L1 and ERM proteins in JEG-3 cells cannot fully mimic the clinical patients with choriocarcinoma received ICB therapy. In addition, the data obtained in this study represents only one type of human choriocarcinoma cell line, despite the existence of genetic and/or phenotypic feature varied from cell lines to cell lines. We should address these issues with more in vitro and in vivo experiments in addition to develop more clinical evidence for better understanding the clinical relationship between PD-L1 and ERM proteins in patients with choriocarcinoma.
Supplementary Materials (Line 30 – 38)
Gene expression profile of programmed cell death ligand-1 (PD-L1), ezrin, radixin, and moesin in human choriocarcinoma cell lines.
Figure S2. Gene expression profile of programmed cell death ligand-1 (PD-L1), ezrin, radixin, and moesin in human choriocarcinoma cell lines. Relative gene expression patterns of PD-L1 in addition to ezrin, radixin, and moesin in three human choriocarcinoma cell lines, JAR, JEG-3, and T3M-3, registered in the database of Cancer Cell Line Encyclopedia (CCLE) were determined by utilizing the Cancer Dependency Map (DepMap) portal data explorer. Scatter plots showing the expression levels (log2 (TPM+1)) of each gene in human choriocarcinoma cell lines. Data from CCLE and DepMap were obtained from the 2021Q3 release.
Reviewer Reply: Author addressed the comment with relevant information to improve the merit of the manuscript.
Comment 2. Please Provide a full blot for all the western Blot data?
Reply Comments.
We would like to appreciate #Reviewer 2’s comment. We have incorporated the original western blotting membrane data as Figure S3 in the Supplementary Materials.
Supplementary Materials (Line 40 – 49)
Original Western Blotting images of programmed cell death ligand-1 (PD-L1), ezrin, radixin, and moesin as well as glyceralde-hyde-3-phosphate dehydrogenase (GAPDH) in JEG-3 cells.
Figure S3. Original Western Blotting images of programmed cell death ligand-1 (PD-L1), ezrin, radixin, and moesin as well as glyceraldehyde-3-phosphate dehydrogenase (GAPDH) in JEG-3 cells. (a) The original Western Blotting membrane to confirm the protein expression of PD-L1, ezrin, radixin, and moesin shown in Figure 1e. (b) The original Western Blotting membrane to detect the protein-protein interaction between PD-L1 and ezrin, radixin as well as actin in the whole cell lysates (input) and those in the immunoprecipitates (IP) using a control antibody or an anti-PD-L1 antibody shown in Figure 4. (c) The original Western Blotting membrane to measure the protein expression levels of ezrin and radixin as well as GAPDH shown in Figure 5c.
Reviewer Reply: Author addressed the comment with relevant information to improve the merit of the manuscript.
Comment 3. Ezrin and Radixin localization in plasma membrane and interaction with PD-L1, thus specific inhibition of ezrin can be a novel strategy to improve ICB therapy. Please provide in-vitro and in-vivo experimental data to support this statement?
Reply Comments.
We would like to appreciate #Reviewer 2’s valuable suggestion. As #Reviewer 2 pointed out, in vitro and in vivo experimental data are absolutely essential to support our statement that specific inhibition of ezrin can be a novel strategy to improve ICB therapy. However, there is no convincing evidence that ezrin function as a scaffold protein to stabilize the plasma membrane localization of PD-L1, leading to the resistance to the current ICB therapy in patients with any cancers. Therefore, we should address these issues with more and more in vitro and in vivo experiments in future studies. With a huge emphasis on the valuable opinions from #Reviewer 2, we mentioned the limitation of this study in the Conclusion section.
Again, we would like to thank your valuable suggestions and constructive comments.
Conclusions (Line 474 – 481)
The limitation of the present study is that the in vitro relationship between PD-L1 and ERM proteins in JEG-3 cells cannot fully mimic the clinical patients with choriocarcinoma received ICB therapy. In addition, the data obtained in this study represents only one type of human choriocarcinoma cell line, despite the existence of genetic and/or phenotypic feature varied from cell lines to cell lines. We should address these issues with more in vitro and in vivo experiments in addition to develop more clinical evidence for better understanding the clinical relationship between PD-L1 and ERM proteins in patients with choriocarcinoma.
Reviewer Reply: Author addressed the comment sufficiently.